# Probing the Efficiency of 13-Pyridylalkyl Berberine Derivatives to Human Telomeric G-Quadruplexes Binding: Spectroscopic, Solid State and In Silico Analysis

**DOI:** 10.3390/ijms232214061

**Published:** 2022-11-14

**Authors:** Carla Bazzicalupi, Alessandro Bonardi, Tarita Biver, Marta Ferraroni, Francesco Papi, Matteo Savastano, Paolo Lombardi, Paola Gratteri

**Affiliations:** 1Department of Chemistry, University of Florence, Via della Lastruccia 3, 50019 Firenze, Italy; 2Laboratory of Molecular Modeling Cheminformatics & QSAR, Department NEUROFARBA—Pharmaceutical and Nutraceutical Section, University of Firenze, Via Ugo Schiff 6, 50019 Firenze, Italy; 3Department of Chemistry and Industrial Chemistry, University of Pisa, Via G. Moruzzi 13, 56124 Pisa, Italy; 4Naxospharma srl, Via G. Di Vittorio 70, Novate Milanese, 20026 Milano, Italy

**Keywords:** G-quadruplex, spectrophotometric titrations, X-ray diffraction analysis, molecular dynamics (MD)

## Abstract

The interaction between the series of berberine derivatives **1**–**5** (NAX071, NAX120, NAX075, NAX077 and NAX079) and human telomeric G-quadruplexes (G4), which are able to inhibit the Telomerase enzyme’s activity in malignant cells, was investigated. The derivatives bear a pyridine moiety connected by a hydrocarbon linker of varying length (*n* = 1–5, with *n* number of aliphatic carbon atoms) to the C13 position of the parent berberine. As for the G4s, both bimolecular 5′-TAGGGTTAGGGT-3′ (Tel12) and monomolecular 5′-TAGGGTTAGGGTTAGGGTTAGGG-3′ (Tel23) DNA oligonucleotides were considered. Spectrophotometric titrations, melting tests, X-ray diffraction solid state analysis and in silico molecular dynamics (MD) simulations were used to describe the different systems. The results were compared in search of structure–activity relationships. The analysis pointed out the formation of 1:1 complexes between Tel12 and all ligands, whereas both 1:1 and 2:1 ligand/G4 stoichiometries were found for the adduct formed by NAX071 (*n* = 1). Tel12, with tetrads free from the hindrance by the loop, showed a higher affinity. The details of the different binding geometries were discussed, highlighting the importance of H-bonds given by the berberine benzodioxole group and a correlation between the strength of binding and the hydrocarbon linker length. Theoretical (MD) and experimental (X-ray) structural studies evidence the possibility for the berberine core to interact with one or both G4 strands, depending on the constraints given by the linker length, thus affecting the G4 stabilization effect.

## 1. Introduction

G-quadruplexes (G4s) are four-stranded secondary structures consisting of a core of multiple stacked ‘G-quartets’, also named guanine tetrads or simply tetrads, which are planar arrays of four guanines involving their Hoogsteen and Watson–Crick faces in eight hydrogen bonds overall [1,2]. G4s are formed by single or multiple polynucleotide chains which have a rich content of contiguous guanines (G-runs or G-tracts), separated by loops which contain a variable number of nucleotides, usually other than guanines. The G4 structures are highly polymorphic and influenced by a combination of factors such as nucleic acid type, loop length, sequence and direction, as well as number and length of G-tracts. In addition, environmental conditions can be crucial in defining the assumed overall folding (molecular crowding, pH, concentration and type of dissolved ions) [3].

Biologically relevant G-quadruplexes were first discovered in eukaryotic chromosomal telomeric DNA [4,5]. After that, pieces of evidence for the in-cell formation of these secondary structures have accumulated [6,7] and numerous functions have been reported, especially regulation of translation in case of RNA [8,9,10] and transcription efficiency, telomere maintenance, regulation of replication/genome stability and CpG hypomethylation for DNA [11,12,13,14,15,16]. Links between G-quadruplexes and disease pathways have recently been reported [17,18]. All of the above underline the importance of this non-canonical structure and emphasize the need for developing tools, including small molecules, with therapeutic or even detection aims [19].

Known G4-targeting ligands generally contain expanded π-systems and/or exhibit high levels of planarity to overlap with the G-quartet structures [20,21,22]. Nevertheless, while these stereochemical features quite easily ensure quadruplex vs. duplex selectivity [23], the recognition of a peculiar G4 geometry may need a much more subtle tuning, even when the expected “sitting-atop” binding mode is in action. Additionally, recent analyses of the interaction between extended condensed aromatic species and different G4 motifs showed that binding modes other than the “sitting-atop” mode may also be at play [24,25]. However, a rational drug design process to develop molecules with different affinity towards different quadruplexes, or with improved binding ability towards a given quadruplex sequence, is still difficult [26]. G4s are difficult targets because of the following: (i) the main interaction often involves stacking with the external guanine quartets, and this structural motif is always present, regardless of the type of sequence and assumed folding; and (ii) despite the considerable interest in this research sector, the number of structures obtained is still limited, especially when compared with the very high number of identified Putative Quadruplex Sequence (PQSs—estimated ca. 376,000 in the human genome) [27,28], and fine structural details of the ligand/target binding are scarce. In fact, to the best of our knowledge, just over 70 ligand/DNA-G4 structures have been deposited into the PDB to date. Notably, in most deposited ligand/G4 structures, the nucleic acid adopts the so-called parallel folding, with propeller loops and external tetrads available for binding. This folding has been repeatedly found for the human telomeric (HT) DNA sequence in the solid state, but also for RNA and c-Myc and c-kit DNA, both in solution and in a solid state.

On the other hand, monomolecular quadruplexes from HT-DNA are known to be highly polymorphic. In solution, they assume different foldings depending on the experimental conditions adopted [1,2,3]. This feature significantly complicates a rational drug-design process, not only because the exact folding present in vivo is not known, but also because it is not clear how, for a given folding, the possible presence of loops and flanking residues will influence the binding. At the state of the art, only seven solution state structures have been solved via NMR spectroscopy for the HT monomolecular quadruplex in complex with ligands; unfortunately, this is still too few to be able to draw definitive conclusions.

Some of us have, for about ten years, been studying the interaction of the HT-G4 with a series of different ligands, comprising metal complexes, natural alkaloids and their derivatives. Our efforts have always been devoted to connecting the stability and solution behaviour of the formed adduct with the structural information obtained from modeling and solid state studies. Among the studied compounds, the natural alkaloid berberine and its derivatives attracted much of our interest.

Berberine (Figure 1) is an isoquinoline alkaloid with a wide range of biological activities [29,30], including anticancer activity [31,32], with relatively low cytotoxicity in healthy cells [33,34]. It has been shown to bind HT-G4, possibly resulting in the inhibition of telomerase, and promoter G4s of human oncogenes, including MYC [35,36].

Several semisynthetic derivatives have been proposed that led to enhanced binding properties and high selectivity towards telomeric G4 DNA. Some crystal structures were determined, evidencing a possible preferential binding mode, with its concave sides in the centre of the G-tetrad [37,38]. Notably, the recently published solution NMR structure of berberine, complexed with the oncogene RET G4-DNA, confirmed this mode of binding [39].

Over the past years, some of us have worked on the chemistry and the DNA binding efficacy studied by employing spectroscopic, calorimetric and molecular modeling techniques, as well as apoptotic properties in relevant cancer cells of the instant series of 13-pyridylalkyl berberine derivatives [40]. Additionally, and more recently, we reported the binding properties towards telomeric G-quadruplexes studied in solution and in the solid state of the berberine derivative, presenting the pyridine moiety connected to the C-13 of berberine with a linker of three carbon atoms (**3**—Figure 1) [41]. In order to further develop this research line, we investigated other berberine derivatives of the same family, characterized by linkers of different lengths. As in previous studies, we focused our attention on connecting structural aspects and binding behaviour.

## 2. Results and Discussion

### 2.1. Solution Studies

The UV-vis signals of the target NAX undergo significant and abrupt changes when increasing amounts of G4 are added and the spectrum undergoes a bathochromic shift (Figure 1a and Appendix A and Table 1). The two maxima initially peaked at ca. 340 and 414 nm, and moved to ca. 348 and 426 nm by interacting with Tel12 or to ca. 342 and 424 nm with Tel23. This occurred for all of the studied systems, which all showed the same spectroscopic features, with the exception of **1**, for which different signal changes were observed with the presence of low-resolved signals for **1**/Tel23 and the generation of a new maximum in the visible region for **1**/Tel12 (Figure 1b and Table 1). In the case of **2**, for both Tel12 and Tel23, the signal changes were more gradual. On the whole, this behaviour indicates the binding of the NAX targets to both G4s, with **2** showing the lowest affinity with respect to a homogeneous behavior of the **2**–**5** compounds, and **1** producing, under the same experimental conditions, a somewhat different adduct. Note: Tel12 is a bimolecular G-quadruplex; throughout the text, concentrations and stoichiometries will refer to its bimolecular form.

The titrations were fitted by means of the HypSpec2014^®^ software (Appendix A) [42]. The software enables, using a least square procedure, the fitting of the experimental UV-vis data over a wide wavelength range, according to different type of models related to multiple equilibria at different stoichiometric ratios (see methods). With the only exception of **1**/Tel12 system, it confirms that a 1:1 binding reaction is sufficient to reproduce the experimental data for the systems investigated here (Appendix A), and provides the values for the equilibrium constants reported in Table 1. Interestingly, in the case of **1**/Tel12, a model which only considers formation of a 1:1 complex is not able to fit the experimental data. In this case, the 1:2 adduct ligand:G4 needs to be added to the 1:1 species in the calculations to reach convergence (Appendix A). The equilibrium relevant to 1:2 complex formation (as **1**/Tel12 + Tel12 ⇆ **1**(Tel12)_2_) is Log*K*′ = 4.0 ± 0.2. Melting studies for different ligand/G4 systems were also conducted at [ligand]/[Tel] ≈ 1 (Figure 2 and Table 1).

On the whole, the solution studies indicate that all systems do strongly bind to G4s, showing a significantly higher affinity towards the bimolecular Tel12 quadruplex (1 to 2 log units higher binding constants), which they stabilize to a much higher extent (approximately 9–16 °C Δ*T_m_* range for Tel12 vs. 2–6 °C range for Tel23). This behavior can likely be ascribed to the presence, in the bimolecular Tel12 G-quadruplex, of an upper tetrad free from the hindrance of the loops. Within the same G4, the behavior of **4** is very similar to that of **5** (with **5** being slightly better than **4**), whereas in the case of **2,** the affinity is significantly lower. In addition, **1** behaves differently from its cognate compounds. This result may be discussed in the frame of flexibility: once the berberine core is anchored on the tetrad, a longer alkyl chain enables the pyridine residue to allocate, at best, either in the tetrad plane or in the grooves. More geometrical details will be obtained by the following studies.

### 2.2. Solid State Study

Crystallization trials were carried out for the ligands **1**, **2**, **4** and **5** in complex with both of the telomeric DNA sequences. Only **2**/Tel12 and **4**/Tel12 adducts produced crystals suitable for diffraction analysis. Both systems crystallize in the tetragonal system, space group P4_2_2_1_2. The two structures are isomorphous and show the adducts in a 1:1 stoichiometric ratio. Overall, comparing the quadruplexes in the two structures, the conformations are very similar to each other, with a RMSD of only 0.38 Å, which was evaluated by superposing all the guanines and the T1-A2 residues (Appendix A). As shown in Figure 3 for **4**/Tel12 and Appendix A for **2**/Tel12, the crystal packing is defined by columns of quadruplex units. The propeller TTA loops determine the junctions between adjacent columns. The two dodecameric chains in each quadruplex are symmetry-related by the four-fold screw axis, and the K^+^ ions in the internal channel are located on this axis or spread over disordered positions in the axis proximity. The quadruplex adopts the expected bimolecular propeller conformation with four stacked quartets overall: three guanine quartets and a TATA quartet located at the 5′-end. This last quartet is built from the 5′-TA ends of the two strands in each quadruplex, and adopts a conformation very similar to what was found in several crystal structures of bimolecular human telomeric G4 [37,41,43,44].

The quadruplexes follow each other along the c axis in a head-to-tail fashion, defining the ligand binding site at each 3′-GGGG/5′-TATA interface. The quartets in direct contact with the ligand, i.e., the 3′-end GGGG tetrad from one quadruplex and the 5′-end TATA from the other, are the most corrugated, with base/base dihedral angles of 12.0° and 16.4° (3′-end GGGG) and 12.7° and 14.6° (5′-end TATA) for **4**/Tel12, or 13.3° and 13.9° (3′-end GGGG) and 3.4° and 11.8° (5′-end TATA) for **2**/Tel12. The two internal guanine tetrads are flatter, and they have base/base dihedral angles in the range of 2.8–9.2° (**4**/Tel12) or 2.7–9.3° (**2**/Tel12).

The A8 residues in the TTA connecting loops are disordered and have been localized in two different positions treated with an occupancy factor of 0.5. As a consequence, each TTA loop adopts two different conformations, which can be described as type-1 and type-3 loops, by using the classification given by Neidle et al. [45]. The type-1 loops are more frequently encountered, and show face-to-face π-stacking interaction between the T6 and the A8 residues (Appendix A).

The ligand **4** is lodged in the binding site at approximately a 3.4 Å π-stacking distance from both the GGGG and TATA tetrads. Figure 4a,b show the details of the contacts established by **4** in the binding site.

The berberine core gives π-stacking with the A2 adenine at the 5′-end, as well as with G5 and G11 at the 3′-end. The charged nitrogen atom points toward the groove, while the functionalized carbon points toward the central channel, so that the aliphatic pendant stretches and locates the pyridine ring just in contact with the second symmetry-related adenine on one side, and with the symmetry-related G5 on the other. It must be underlined that, from a crystallographic point of view, the two adenines, on one hand, and the symmetry-related G5 and G11 residues, on the other, only appear to behave in different ways. In fact, the four-fold screw axis relating the two DNA dodecamers also applies to the ligand molecule, and the overall electron density map in the binding site clearly shows two symmetry-related ligand molecules sharing the same position with half occupancy factors (Appendix A), both pointing the pendant alkyl arm above the central quadruplex channel.

As shown in Figure 4a,b, **4** is involved in additional water-bridged H-bond contacts with the DNA polymer. The HOH-2 molecule bridges a C–H group at a long distance from pyridine (4.0 Å), and the phosphoester (2.9 Å) and ribose (3.0 Å) oxygens from T1, while the HOH-1 water molecule bridges a G5 phosphate oxygen (2.5 Å) and one oxygen from the benzodioxole group of the berberine core (2.3 Å). Notably, water-bridged or direct DNA/ligand H-bonds involving benzodioxole oxygen were also found in the crystal structure of the **3**/Tel12 adduct [41], as well as of the adduct formed with Tel12 by the berberine cognate compound coptisine [43]. This is evidence of an important role played by this group in stabilizing these alkaloids in the biomolecule binding site (Appendix A).

In addition to the H-bonds given by the benzodioxole group, the **4** structure confirms the structural features previously evidenced by the crystal structures of berberine and other C-13 berberine derivatives with Tel23 [37,38] and Tel12 [41]. As a matter of fact, in all these structures, the berberine is placed on top of two lateral guanine residues, with its cationic nitrogen far away from the partial negative charges featuring the carbonyl oxygens in the quadruplex central channel. These findings may indicate some preferential requirements for the binding of the berberine core to the telomeric quadruplex structure. In that position, the C13 carbon atom points toward the inner potassium channel, and the pendant alkyl chains are properly arranged to place the aromatic group in stacking contact with the bases of the binding site (Appendix A).

As far as the **2**/Tel12 adduct is concerned, the binding site is again defined by the 3′-end GGGG and 5′-end TATA tetrads from subsequent units in a column, and the **2** molecule is placed at stacking distance from each, at about 3.5 Å. Details of the interaction are shown in Figure 4c,d. As already seen for **4**, two molecules of **2** treated with occupancy factor 0.5, which are symmetry-related by the screw axis parallel to the quadruplexes’ column, account for the apparent difference in behavior of the symmetry-related residues in the binding site (Appendix A).

Notably, the ligand occupies a somewhat different position in the binding site with respect to the cognate compound **4**, but also with respect to parent berberine and other C-13 derivatives. As shown in Figure 4c,d, the NAX120 molecule still presents the aliphatic pendant pointing toward the center of both tetrads, but the berberine core is diagonally placed with respect to the quartet, being in contact with residues symmetry-related by the 4-fold screw axis: the two adenines in the 5′-end TATA tetrad (Figure 4c) and the two G5 guanines in the 3′-end GGGG tetrad (Figure 4d). The pyridine ring is placed near T1 and G11.

The different binding mode shown by **2** may be due to the shorter aliphatic chain, which features only two carbon atoms. It is easy to suppose that moving the positively charged berberine core from the periphery to the center would be more favorable than placing the pyridine ring above the tetrad central channel, as it features the excess of electron density of the carbonyl oxygens. This is the first evidence of a possible adaptive binding of the berberine core which has taken into account pendants’ structural and electronic requirements.

### 2.3. Hirshfeld Surface’s Analysis

In order to obtain a direct visual comparison of the interactions established by the three berberine derivatives, **2**, **3** and **4**, with the guanine tetrad of Tel12, we applied to their crystal structures a modified version of the Hirshfeld surface’s analysis method (please see Appendix A for details). Without giving further details, for which we refer to the specific literature, we can say that the Hirshfeld surface of a specific molecule in a given crystal packing is related to its van der Waals surface. This method takes into consideration neighbouring molecules; hence, it is useful to study the intermolecular interactions [46,47,48,49].

Results are shown in Figure 5, where contacts are shown as function of colors, from blue (longer than vdW contacts) to red (shorter than vdW contacts); white represents contacts equal to vdW ones.

Interestingly, the calculated surfaces are in very good agreement with the results of the solution study for the dodecameric sequence. Actually, most of the surface of **2** (Figure 5a) is white, signaling a lack of specific contacts, with the exception of a single red spot in proximity to the out-of-plane ethylenic group in the berberine core, merely because it is closer to the plane of the GGGG tetrad. Several dark red spots are localized on the surface of **4** in proximity to G5 and to the HOH-1 water molecule (Figure 5c), indicating strong and localized interactions. On the other hand, the surface of **3** is characterized by lighter red spots, indicating somewhat weaker interactions, spread all over the surface and significantly involving 3 out of 4 guanines (Figure 5b). In fact, the higher Δ*T_m_* value observed in UV-melting experiments for the **3**/Tel12 adduct, with respect to the **4**/Tel12 complex, led us to infer that **3** is able to interact with both strands of the bimolecular quadruplex more effectively than **4** does, despite the slightly lower binding affinity evidenced by the Log*K* values (Table 1).

### 2.4. Molecular Modeling Study

All trials which attempted to crystallize the ligands with 23-mer human telomeric sequences were ineffective, precluding us from paralleling the structural investigation carried out for the ligand/Tel12 adducts. Therefore, we performed molecular dynamic simulations based on the empirical force fields method to analyze the possible binding mode of the tested ligands with the mononuclear G4. This approach has the advantage that different foldings can be considered, as well as the possible influence of an aqueous environment. As is known, these methods are able to manage the G4 structures’ size and, because no covalent bonds are broken and/or formed, they are a suitable choice as long as the used force field has been properly chosen [50]. To this aim, we chose the OPLS4 force field, which has been proven to provide reliable results [51].

We verified the binding towards the propeller folding, typical of the solid state and the type 1 hybrid structure, which is known to be assumed by the wild type telomeric sequence in potassium-containing dilute solution [52].

The structures of the adducts formed with the propeller G-quadruplex are shown in Appendix A. Overall, these results agree with the binding mode evidenced for the Tel12 sequence, thus confirming the accuracy of the method. With the four tested ligands, in all but one of the orientations, we found the berberine core placed along a side of the quadruplex, in π-stacking contact with two guanine residues and with the pendant arm pointing towards the center of the tetrad. The pyridine ring is often involved in π-stacking interaction as well.

Calculated models for the adducts formed with the type 1 hybrid structure agree with the results obtained by solution study. The structures are shown in Figure 6, and Table 2 summarizes the most important ligand/target interactions. Each NAX ligand forms an adduct, with the human telomeric DNA sequence adopting the 3 + 1 hybrid-1 G-quadruplex folding. In addition, **2** is confirmed to be the less interacting ligand, while **3**, **4** and **5** resemble each other in terms of type and strength of contacts.

However, while **4** and **5** use both the berberine core and the pendant pyridine ring to establish strong π-stacking interactions with the DNA, **3** seems to use only the berberine core, and the strongest interactions appear to be limited to the contiguous A3-G4 residues. This finding could be in agreement with the low melting temperature measured for **3** with respect to **4** and **5** (Table 1).

An additional aspect evidenced by this modeling result is the possibility for the benzodioxole group to be involved in H-bond interactions with the NH group from adenine (**2** and **4**) or thymine (**5**) residues.

## 3. Materials and Methods

### 3.1. Materials

Ligands **1**–**5** (Figure 1) were received by Naxospharma srl and used without further purification. As previously reported [40], 13-pyridylalkyl berberine derivatives were synthesized from berberine chloride, and the appropriate pyridine (alkyl)carboxaldehyde (Figure 2), via a modification of an unusual enamine–aldehyde condensation, performed on 7,8-dihydroberberine [53].

DNA oligonucleotides 5′-TAGGGTTAGGGTTAGGGTTAGGG-3′ (Tel23) and 5′-TAGGGTTAGGGT-3′ (Tel12) were purchased from Metabion (both HPLC purity). Stock solutions were prepared in 0.1 M KCl (Sigma, Milano, Italy) + NaCacodylate 2.5 mM (sodium cacodylate, (CH_3_)_2_AsO_2_Na, pH = 7.0 buffer, Sigma) and their molar concentration (always expressed as G4s) was calculated according to the weight/content provided by the sample certificates. Tel12 forms bimolecular G4s: the molar concentration in G4 is half that in strands. Throughout the text, Tel12 concentrations and stoichiometries are expressed with respect to the biomolecular structure. The formation of the G4 structure is carried out by slow heating up to 90 °C for 10 min and slow cooling down to room temperature. The solutions were then stored overnight at 4 °C. MilliQ water (crystallization experiments) or ultra-pure grade water from a SARTORIUS Arium-pro water purification system was used as the reaction medium. All reactants not specifically mentioned were analytical grade and were used without further purifications.

### 3.2. Solution Studies

A Shimadzu UV-2450 spectrophotometer was used for spectrophotometric titrations and melting experiments. The apparatus was equipped with a Peltier temperature controller to maintain the temperature within ±0.1 °C of the setting value. In the spectrophotometric titrations, increasing amounts of the G4 were added directly into the cuvette containing the target compound (**1**–**5**, Figure 1); the spectra were recorded at a slow (60 nm/min) scan rate. The spectrophotometric data were analyzed by means of the HypSpec2014^®^ software [42], which first uploads the whole spectral range for each point of the titration and the data on G-quadruplex and ligand concentrations for each spectrum, then fits the absorbance changes on proposed models according to multiple equilibria of different stoichiometries (see legend of Appendix A). Melting experiments were carried out by heating the working solutions from 25 °C to 90 °C at a scan rate of 5 °C/min, each 6.5-min step being composed of 4.5 min rest, 1 min spectrum recording and 1 min temperature increase. In these experiments, [ligand] ≈ [G-quadruplex] = 1–2 × 10^−5^ M, buffer is KCl 0.1 M, NaCac 2.5 mM and pH is 7.0. The lines correspond to the sigmoidal fits of the experimental trends, whose inflection point corresponds to *T_m_*. Melting curves refer to absorbance signal changes with temperature at 295 nm [54] using a 2 mm path-length cuvette; melting temperature (*T_m_*) is obtained as the inflection of the sigmoidal fit of the trend. Titrations and melting experiments were performed in triplicate; errors are ±SD over the triplicates.

### 3.3. X-ray Diffraction Analysis—Crystallization

Crystallization screenings were performed for the human telomeric sequences Tel12 and Tel23 as well as the ligands **1**, **2**, **4** and **5 [55]**. The **3**/telomeric G4 system was already subjected to crystallization experiments, and the obtained structure was previously reported [41]. Crystallization experiments were set up by using the sitting drop crystallization method. Suitable crystals for X-ray analysis were obtained for the adducts formed by Tel12 and **2** or **4** by mixing 1 μL of 0.001 M G4 DNA/ligand 1:1 adduct with 1 μL of crystallizing solution. When the ligand was **2**, a reservoir (100 μL) was filled with the crystallizing solution, which contained 0.05 M NaCacodylate with pH = 6.5, 0.1 M NaI and 35% *v*/*v* 2-methyl-2,4-pentanediol (MPD). When the ligand was **4**, the crystallizing solution contained 0.05 M KCacodylate with pH 6.5, 0.1 M Li_2_SO_4_ and 20% *v*/*v* PEG400, and the reservoir contained a 30% *v*/*v* PEG400 aqueous solution (100 μL).

### 3.4. X-ray Diffraction Analysis—Data Collection and Structure Solution and Refinement

Diffraction experiments on the crystals were performed at ESRF-ID30B beamline, lowering the temperature to 100 K and using the crystallization condition as cryoprotectant solution either as such (**2**) or containing 40% *v*/*v* PEG400 (**4**). Data were collected up to a maximum resolution of 2.0 (**2**) and 1.6 Å (**4**), using a 0.9726 Å wavelength X-ray. Data were integrated and scaled using the program XDS [56].

The structures of the adducts were solved by molecular replacement using the program Phaser1.14 [57]. The crystallographic coordinates of the Tel12 binuclear quadruplex adduct (PDB accession number 5CDB) [37] were used as a search model, after deleting atoms belonging to ligand and solvent molecules. The Fo-Fc electron density map of each structure showed a clear density for the drug molecule located on the 4-fold screw axis. Both models were refined with isotropic thermal factors using the program Refmac5.8.0151 from the CCP4 package [58,59]. Geometrical restraints for **2** and **4** were generated by using the Grade web server [60], and the ligand molecules were added manually into the proper electronic density maps. Manual rebuilding of the models was performed using the program Coot [61]. Each structure shows disordered TTA loops, resulting in overall poor electron density maps. The A8 residue has been localized in two different positions with occupancy factor 0.5. For the base atoms of the T6 residue of **4**/Tel12 0.0, an occupancy factor was used, as they were not clearly localized in the map. The described disorder is most likely responsible for the tetragonal symmetry found, which can be interpreted as a pseudo-symmetry phenomenon. Nevertheless, all attempts carried out to treat these data with different cells and/or lower symmetry space groups showed no results. The crystal packing analysis was conducted by means of the Mercury program [62]. Final coordinates and structure factors have been deposited into the Protein Data Bank (PDB accession number 8AAD for **2**/Tel12 and 7ZUR for **4**/Tel12). Statistics on the data collection and refinement are reported in Appendix A. Figures of the structures were generated with the UCSF Chimera package [63]. Hirshfeld-like surfaces, i.e., maps of normalized contact distance (d_norm_) plotted on promolecule surfaces, were generated using the software CrystalExplorer v17 to help with relevant contact analysis [64].

### 3.5. In Silico Studies

The crystal structure (PDB: 5MVB) [65] used for computational studies was prepared using the Protein Preparation Wizard tool implemented in the Schrödinger suite, assigning bond orders, adding hydrogens, deleting water molecules and optimizing H-bonding networks. An energy minimization protocol with a Root Mean Square Deviation (RMSD) value of 0.30 Å was applied using an Optimized Potentials for Liquid Simulation (OPLS4) force field [51,66,67,68].

The 3D ligand structures were prepared by Maestro (v.12.9) [66] and evaluated for their ionization states at pH 7.4 ± 0.5 with Epik (v.5.7) [66]. The conjugate gradient method in Macromodel (v.13.3) [66] was used for energy minimization (maximum iteration number: 2500; convergence criterion: 0.05 kcal/mol/Å^2^).

For molecular docking studies, the software Glide SP (v.9.2; default settings) [66] was used. In this regard, grids were centered in the centroid of the complexed ligand. The standard precision (SP) mode of the Glide Score function was applied to evaluate the predicted binding poses.

Molecular dynamics (MD) simulations were performed using the Desmond Molecular Dynamics System (v.6.7) [66] (Schrödinger suite) and OPLS4 force field. All systems were solvated in an orthorhombic box using simple point charge water molecules, which were extended 15 Å away from any DNA atom. The simulation protocol included a starting relaxation step followed by a final production phase of 100 ns. In particular, the relaxation step comprised the following: (a) a stage of 100 ps at 10 K retaining the harmonic restraints on the solute heavy atoms (force constant of 50 kcal/mol/Å^2^), using the NPT ensemble with Brownian dynamics; (b) a stage of 12 ps at 10 K with harmonic restraints on the solute heavy atoms (force constant of 50 kcal/mol/Å^2^), using the NVT ensemble and Berendsen thermostat; (c) a stage of 12 ps at 10 K and 1 atm, retaining the harmonic restraints and using the NPT ensemble and the Berendsen thermostat and barostat; (f) a stage of 12 ps at 300 K and 1 atm, retaining the harmonic restraints and using the NPT ensemble and the Berendsen thermostat and barostat; (g) a final 24-ps stage at 300 K and 1 atm without harmonic restraints, using the NPT Berendsen thermostat and barostat. The final production phase of MD was run by using a canonical NPT Berendsen ensemble at 300 K. During the MD simulation, a time step of 2 fs was used while constraining the bond lengths of H atoms with the M-SHAKE algorithm. The atomic coordinates of the system were saved every 100 ps along the MD trajectory. Protein RMSD, ligand RMSD/RMSF (Root Mean Square Fluctuation) ligand torsions evolution and occurrence of intermolecular H-bonds and hydrophobic contacts were provided by the Simulation Interaction Diagram (SID) implemented in Maestro, along with the production phase of the MD simulation. The tool reads the MD trajectory file and identifies ligand/target interactions which repeatedly occur during the simulation time (for instance, a 60% value suggests that the interaction is maintained for the 60% of the MD). The 1000 frames resulting from MDs were clustered using the Conformer Cluster tool implemented in the Schrödinger suite in 10 clusters. The representative poses of the most abundant clusters were refined with Prime MM-GBSA calculations (v.5.5) [66] using a VSGB (Variable Surface Generalized Born) solvation model, considering the target to be flexible within 3 Å around the ligand. Figures were generated using Chimera and Maestro [63,66].

## 4. Conclusions

Intercalation in the G4s rigid structure is quite rare, whereas stacking on the external tetrads is more probable [69]. Interaction data for the human telomeric DNA with the NAX series, obtained from both solution and solid state experiments, confirm this statement. It might be speculated that the sitting atop binding mode occurring in the solid state could be replicated in solution in the case of Tel12 bimolecular quadruplex, where the tetrad is totally free and available to bind the probe. The steric hindrance by the loops drives the lower binding constants by Tel23 with respect to Tel12 adducts. The Log*K*s measured from UV-vis titrations actually demonstrate a high affinity for the studied ligands for Tel12, as expected when optimal π–π stacking occurs [70]. Apart from **1**, which shows a different stoichiometry ratio, for the other ligands, similar behaviors can be inferred by their spectral profiles. Notably, an interesting correlation between strength of binding and length of the alkyl chains can be inferred on the basis of the theoretical (in silico) and experimental (X-ray) results obtained: the optimal interaction of the pyridyl pendant in 13 with the tetrad is reached when the linker is 4–5 carbon atoms long, regardless by the parallel or hybrid folding of the quadruplex. For Tel12, the lateral residues are less important, and the berberine derivative, together with its pending arm, is located on the tetrad.

In the case of adducts formed with Tel23, the binding is still favorable. Although the tetrad is not completely free now, the overall data allow us to suppose that the stacking contribution from berberine core still predominates, confirming the binding preference with the pendant harm pointing towards the center. Nevertheless, the involvement of loops and/or flanking residues in the binding process is confirmed to be able to finely tune the overall strength of the adduct, thus resulting in a stability trend which is more difficult to interpret. This aspect may warrant interest. It is known that the binding features are the complicate superimposition of interactions between tetrads and loops, whose relative strength comes from small details in both host and guest structures; on the other hand, these loop interactions are those which may determine selective binding to a peculiar sequence of biomedical interest [26].

Despite being less pronounced, the correlation between strength of binding and length of the alkyl chains is still visible, and strictly recalls what was reported in the previous literature about C-13 substituted berberine derivatives [37]. The point of linker length may produce interesting results, for instance, in the frame of dimeric DNA quadruplexes with unique biological functions [71]. It is also known that the presence of substituents at different distances from the berberine core may improve the binding selectivity and tune cellular effects [35]. In this work, we have shown through different experimental approaches the details of the binding features as well as a very different affinity between monomolecular (Tel23) and bimolecular (Tel12) G-quadruplex. In that sense, selectivity was achieved. The prosecution of this work may be to further inspect the ability to discriminate different geometries, as well as comparing longer linkers, which may be allocated into the grooves.

## Data Availability

Not applicable.

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
