# Peer review of "Probing the Efficiency of 13-Pyridylalkyl Berberine Derivatives to Human Telomeric G-Quadruplexes Binding: Spectroscopic, Solid State and In Silico Analysis"

_ijms, 2022, doi:10.3390/ijms232214061_

Round 1

Reviewer 1 Report

This work aims at probing the efficiency of 13-pyridylalkyl berverine derivatives to Human Telomeric G-quadruplexes binding by means of different experimental techniques and computational methods. Five berverine derivatives, in which the alkyl chain in position C13 increases systematically from n=1 to n=5 and two GQ sequences where considered (Tel12) and (Tel23), having Tel12 better affinity. The authors highlight not only the importance of the H-bonds coming from benzodioxole in the interaction but also the correlation between the strenght of the interaction and the hydrocarbon linker length.

This reviewer considers that the topic of research, use of GQ as therapeutic targets and their interaction with small molecules is a hot topic of research in the field of nanobiotechnology, biochemistry, biomedicine and other scientific disciplines and therefore the present work is not only interesting for the readers of this journal but also timely and could help to achieve new challenges in this topic of research. Moreover, the authors seem to achieve some structures that they put in the PDB, which also enrich their work.

The bibliography includes very recent citations (2018 - currently) along with older references (early 2000 and 80's). Nevertheless, there are very recent works on the affinity/selectivity of small molecules with GQ and on the computational treatment of such systems as the following PCCP publications: https://doi.org/10.1039/D2CP02241A or https://doi.org/10.1039/D2CP00214K that would improve the bibliography of the Introduction and have to be discussed in the introduction. Moreover, recent reviews on the topic: Ann. Rev. Biophys. 2021, 50, 209 and Molecules 2021, 26(16), 4737 have to be cited and some references therein would give a better perspective on the state-of-the-art of this interesting topic of research.

When the authors discuss the hydrogen bonds and weak interactions, this referee is a little bit confused about how they consider the hydrogen bonds and weak interactions. Are the authors using just geometrical criteria ? Hirshfeld surfaces give quantitative or qualitative information ? Perhaps some QTAIM analysis including values for the electronic density in bond critical points could help to give a quantitative description of the hydrogen bonds and weak interactions. Of course, not for all the systems and the whole system but for the smallest and largest system by taking into account some reduced model for the system could help to give some quantitative idea of the forces of the hydrogen bonds and the weak interactions.

At some point, the authors talk about pi-pi stacking between the berverine derivatives and the DNA GQ. This is an interesting point considering that the berverine derivatives are charged positively. pi-pi stacking between small molecules and DNA is ruled usually by dispersion, see PCCP 19 (2017) 16638, JACS 124 (2002) 3366, JCIM 59 (2019) 3989 or RSC Adv. 11 (2021) 1553. However, in some cases the electrostatic contribution becomes the most important interaction between the small molecule and the DNA despite to the pi-pi stacking, see JCC 43 (2022) 804 or Mutat. Res.-Fund. Mol. M. 2007, 623, 72. Because of it, some analysis of the interaction energy by means of some of the different schemes in the bibliography (any kind of EDA, SAPT, etc.) would be very interesting and useful to describe the nature of the forces that rule the interaction. Again, this calculations have to be performed just for some systems and the authors could also use reduced models of the chosen systems.

About the docking studies, do the docking studies led to other modes of interaction of the small molecules with the GQ ? That is groove binding or binding in the loops ? Where they very different in energy compared to the sitting atop or end of stacking mode.

Why the authors chose OPLS4 force field for the simulations ? This force field is adequate to study non-canonical GQ structures ? The authors have to give some references or some calibration calculations that justify the use of such force field for non-canonical GQ structures.

In page 2, line 56 the following sentence looks incomplete: "Nevertheless, while these stereochemical features quite easily ensure quadruplex vs duplex selectivity[23]."

In page 9, line 279 the following text is difficult to understand: "This method is commonly applied to low molecular weight spe-279 cies, the overall structure of which is subjected to the calculation. Nevertheless, consider-280 ing that the Hirshfeld surface of a given molecule is sensitive only to the direct contacts it 281 establishes (in other words local electronic density at any point is function of nearby mol-282 ecules only), in order to reduce the extent of our problem we decided to delete all atoms 283 but the ligands and the 3’-end GGGG tetrad." 

In Figure 4 and Figure S6 of the Supporting Information the authors have to change the colour for the ions since is very similar to the colour for C atoms.

The authors put Kcal/mol in with K in capital letters and it has to be written without capital letters as kcal/mol.

The authors must address all these concerns in a new version of the work.

Author Response

  • The bibliography includes very recent citations (2018 - currently)along with older references (early 2000 and 80's). Nevertheless,there are very recent works on the affinity/selectivity of smallmolecules with GQ and on the computational treatment of suchsystems as the following PCCP publications:https://doi.org/10.1039/D2CP02241A orhttps://doi.org/10.1039/D2CP00214K that would improve thebibliography of the Introduction and have to be discussed in theintroduction. Moreover, recent reviews on the topic: Ann. Rev.Biophys. 2021, 50, 209 and Molecules 2021, 26(16), 4737 haveto be cited and some references therein would give a betterperspective on the state-of-the-art of this interesting topic ofresearch.

We thanks the reviewer for her/his suggestions. More than a theoretical computational investigation on the energetic contributions to the interaction, the study mainly focuses on the in solution characterization and on the structural aspects of the interaction of berberine derivatives with G-quadruplexes (G4). According to an approach typical of many medicinal chemistry studies that often uses empirical forcefields, our investigations mainly focuses on the forces that, overall, drive the stabilization of the complexes formed by the ligands and the G4. The detailed analysis of the electronic structure and energetics in the interaction is out of the aims of the present study. We would like to let this referee know that we had indicated the biochemistry area as preferential for the topics treated in the manuscript.

Nevertheless, we thank the referee for the interesting suggestions which could be at the base of possible future studies more addressed on theoretical aspects. Here, to better clarify the kind of study performed, we modified the discussion section to explicitly mention empirical forcefield methods and we added one of the references suggested by the referee.

Section 2.4. Molecular modelling study

“….Therefore, we performed molecular dynamic simulations based on empirical forcefield’s method to analyse the possible binding mode ….” and then “…As known, these methods are able to manage the G4 structures’ size and because no covalent bonds are broken and/or formed, they are a suitable choice as long as the used force field has been properly chosen [42].”

Ref 42 Ortiz de Luzuriaga, I.; Lopez, X.; Gil, A.; Learning to Model G-Quadruplexes: Current Methods and Perspectives. Annu. Rev. Biophys. 2021. 50:209–43. https://doi.org/10.1146/annurev-biophys-060320-091827

  • When the authors discuss the hydrogen bonds and weak interactions, this referee is a little bit confused about how they consider the hydrogen bonds and weak interactions. Are theauthors using just geometrical criteria ?

The geometrical criteria here usedfor the evaluation of intermolecular contacts are well established and accepted by the scientific community (cfr I. J. Bruno et al. J. Comput.-Aided Mol. Des., 1997, 11 , 525 —537, P. A. Wood et al., CrystEngComm(2013), 15, 65–72)

  • Hirshfeld surfaces give quantitative or qualitative information?

Like all visual representations of crystal structures, Hirshfeld surfaces offer a qualitative depiction of intermolecular interactions. The quantitative information, e.g. contact distance, is expressed with a colour code: this allows to easily detect molecular regions that are closely or loosely interacting. This view is more holistic (regions rather than atoms) with respect to the classic pairwise dash annotated with atom … atom contact distance (already presented in the manuscript), yet ultimately equivalent to it. When the crystal structure is used as a whole, Hirshfeld surface analysis provides additional quantitative information in terms of percentage of atom … atom contacts. This performs particularly well for small molecules’ crystal structures, normally obtained at very high resolution. In our case, due to the lower resolution with respect to small molecules’ structures and to the higher atomic dimensionality, we prefer to use Hirshfeld analysis in just a qualitative, yet easy to see way, which clearly highlights contact regions and the different ability of the ligands to interact with the tetrad.

  • Perhaps some QTAIM analysis including values for the electronic density in bondcritical points could help to give a quantitative description of thehydrogen bonds and weak interactions. ………. Again, thiscalculations have to be performed just for some systems and theauthors could also use reduced models of the chosen systems.

Please, see answer to point 1

  • About the docking studies, do the docking studies led to othermodes of interaction of the small molecules with the GQ ? Thatis groove binding or binding in the loops ? Where they verydifferent in energy compared to the sitting atop or end of stacking mode.

Berberine and berberine derivatives like NAXs are able to bind G4 by stacking on external tetrads. This behaviour was expected both according to literature and our previous experience. Accordingly no ligand poses were achieved interacting with groove. When the target was DNA in hybrid 1 topology, the lignad was found to interact also with some loop or flanking residues. Involved residues are shown in figure 8 and kind of interactions are detailed in Table 2

  • Why the authors chose OPLS4 force field for the simulations ?This force field is adequate to study non-canonical GQ structures? The authors have to give some references or some calibrationcalculations that justify the use of such force field for non-canonical GQ structures

We modified the Section 2.4. Molecular modelling study as follows: “To this aim, we chose the OPLS4 force field, which has proven to provide reliable results [43].”

Ref 43 Lu, C., Wu, C., Ghoreishi, D., Chen, W., Wang, L., Damm, W., Ross, G.A., Dahlgren, M.K., Russell, E., Von Bargen, C.D., Abel, R., Friesner, R.A., Harder, E.D., OPLS4: Improving Force Field Accuracy on Challenging Regimes of Chemical Space. J CHEM THEORY COMPUT, 2021, 17, 4291-4300; https://doi.org/10.1021/acs.jctc.1c00302

Ref 43 in the revised version was already present in the old version as ref 49c.

  • In page 2, line 56 the following sentence looks incomplete:"Nevertheless, while these stereochemical features quite easilyensure quadruplex vs duplex selectivity[23]."

There was a typo in the dot which should be a comma. The sentence has been modified as follows: “…selectivity [23], the recognition…”

  • In page 9, line 279 the following text is difficult to understand:"This method is commonly applied to low molecular weight species, the overall structure of which is subjected to thecalculation. Nevertheless, considering that the Hirshfeldsurface of a given molecule is sensitive only to the directcontacts it establishes (in other words local electronicdensity at any point is function of nearby molecules only), inorder to reduce the extent of our problem we decided to deleteall atoms but the ligands and the 3’-end GGGG tetrad."

Please, see point 18 in Reply to Reviewer 3: “Following the comments of the referee the description of the method has been moved to SI and the Section 2.3 significantly shortened.”

  • In Figure 4 and Figure S6 of the Supporting Information the authors have to change the colour for the ions since is verysimilar to the colour for C atoms

Done

  • The authors put Kcal/mol in with K in capital letters and it has tobe written without capital letters as kcal/mol

Done

Reviewer 2 Report

The Authors report an interesting study of interaction between the series of berberine derivatives with alkyl linkers of various lengths and human telomeric G-quadruplexes (Tel12 and Tel23). Using UV-Vis spectroscopy, they recorded changes in spectra upon binding of ligands and quadruplexes and upon heating solutions of ligand free and ligand bound G-quadruplexes. Crystals of Tel12 G-quadruplex complexes with two ligands were obtained and structures were determined. Authors were unable to obtain usable crystals from Tel23 G-quadruplex:ligands complexes so they used molecular modelling studies to try to interpret results obtained in UV-Vis experiments. Authors report an interesting correlation between strengths of binding and length of the alkyl chains for Tel12 and discuss the involvement of loops and/or flanking residues in the binding process during formation of Tel23 adducts. The manuscript reports some interesting findings, is well-written but there are some major issues/comments to consider before publication:

Line 15: Perhaps authors should mention that they are using berberine derivatives with increasing alkyl chain (or linker) length where n represents number of carbon atoms.

Line 58: comma instead of dot

Line 95: description of Scheme 1 is missing

Figure 1: Very low absorbance. The degree of error is expected to be high at low concentrations. The authors should comment on their decision for such low concentrations. Spectra of individual titrations should be removed since the changes are on the order of the inherent noise in the measurement. According to manufacturer site Shimadzu's UV-2450 photometric accuracy is ± 0.002 Abs in the range 0-0.5 Abs. Resolution is low. I find it interesting that ligand 4 decreases absorbance whereas ligand 1 increases the absorbance.

Table 1: Please refrain from using word »Thermodynamic« when talking only about melting temperatures and equilibrium constants. Also, symbols representing physical quantities or mathematical variables should be written in italic type. How were the error values calculated? Is [Tel12] concentration of single strand or concentration of bimolecular structure? Are all binding stoichiometries calculated to bimolecular structure Tel12?

Line 145: [ligand]/[Tel] = ca. 1: “ca.” is not explained

Figure 2: Why did authors decide to show titration curves at wavelengths of 342 and 349 nm? These values can't be found in table 1 (lmax). Graph with spectra has axis label »Negative absorbance«. Why? Differences in measured absorbances are the same as photometric accuracy (0.002). The obtained equilibrium constants are therefore questionable. What does it mean “% formation relative to B”, specifically what does “B” stand for? Experimental conditions are described as: “Experimental conditions: see Figure 1B and 2”. What do the mean by Figure 1B and 2? This is Figure 2 and there are no experimental conditions. At the bottom of panels A and B, there are some additional graphs which are not described (“Bottom panels are the residuals” is not an appropriate description). The authors are showing absorbance spectrum and relevant deconvolutions. They should also show measured spectra of free ligand to compare it with deconvoluted spectra. Legend should be displayed next to graphs.

Figure 3: Graphs are not aligned. Legends are missing. Red line is not visible. Why are authors not showing melting curve for 3+G4? What kind of model function was used to draw melting curves?

Lines 160-164: This discussion could/should be extended and improved (e.g. correlation between strengths of binding and length of the alkyl chains, differences in equilibrium constants of Tel12 and Tel23…). Also, in some places bold numbers are used to describe ligands (1, 2, 3, …) in other places these numbers are not bold.

Figure 4: Panel labels are different in all figures ((a), (A), A). They should be unified.

Line 193: »The 4 ligand« should be replaced by »The ligand 4«

Figure 5: see previous comment

Line 266: Why were only 2, 3 and 4 selected for Hirshfeld surface’s analysis? I guess the selection was not based on ability of derivatives to form crystals since only 2/Tel12 and 4/Tel12 adducts gave crystals suitable for diffraction analysis. So, what was the criteria for selection of these three derivatives?

Line 306: “The higher DTm value found in solution solution…” This should be rewritten to something similar to; “The higher DTm value observed in UV-melting experiment…”

Line 324: Perhaps “orientation” should be used instead of “poses”?

Line 328: Why »on the other hand«? Are results different or unexpected?

Table 2: Benzodioxole should be written in one line

Line 473: Word »long« should be used instead of »length«.

Line 475: Why »on the other hand«? The sentence »In the case of adducts formed with Tel23, the binding is still favorable. « is enough.

Author Response

  • Line 15: Perhaps authors should mention that they are using berberine derivatives with increasing alkyl chain (or linker) length where n represents number of carbon atoms

Line 18 of the abstract has been modified as follows:

“…hydrocarbon linker of varying length (n = 1- 5, with n number of aliphatic carbon atoms) to the C13 position …”

  • Line 58: comma instead of dot

Done

  • Line 95: description of Scheme 1 is missing

The following caption has been added: “Scheme 1. Drawing of Berberine and its derivatives 1-5”. Please note that Scheme 1 was changed as requested by referee 3.

  • Figure 1: Very low absorbance. The degree of error is expectedto be high at low concentrations. The authors should commenton their decision for such low concentrations. Spectra ofindividual titrations should be removed since the changes are onthe order of the inherent noise in the measurement. According tomanufacturer site Shimadzu's UV-2450 photometric accuracy is ± 0.002 Abs in the range 0-0.5 Abs. Resolution is low. I find itinteresting that ligand 4 decreases absorbance whereas ligand 1increases the absorbance.

The reviewer is perfectly right to raise this point. Unfortunately, these low absorbance conditions were the result of different experimental constraints. We prepared G-quadruplex (G4) solutions at the higher concentrations possible, given the quantities provided, the protocol for the annealing process, and working solutions with the desired buffer. It meant something like 1E-4M for the G4, i.e. the titrant. This value turned into more diluted ones in the measuring spectrophotometric cell (from 0 to ca. 2E-5M). The berberine concentrations needed to be significantly lower than this upper limit, enabling to span over a ratio G4/berberine from zero to G4 excess during the titration. On the whole, this produced diluted berberine starting solutions. However, the experiments were done with maximum care, to balance this unfavourable point. (A) The spectra were registered in slow scan rate, so to enable accumulation (added in the revised text); (B) the experiments were repeated at least in triplicate so to check data reliability (added in the revised text); (C) the absorbance/titrant concentration trends were checked and data dispersion was found low (see the agreement between blue diamonds/experimental and red crosses/calculated in the left panel of Figure 2, the residuals [experimental minus calculated] lye in the order of 1E-3 absorbance units; Figure 2 was moved to SI); (D) data were analysed by software which uses all of the 300 wavelengths uploaded and returns a K value which corresponds to the best fit over all the binding isotherms drawn for each of the wavelengths, such a procedure reducing the errors. Also, note that being aware of the inherent error of the procedure, the authors did not pretend to produce K values with many digits. The numbers in Table 1 are reported as LogK and the error on them is at least ± 0.2 log units, which means a somewhat high error on K numerical values. Being aware of all the above and having (duly) reported a significant error in logK values, the differences in the experimental/spectral behaviour and Table 1 still show that some comment on the trends of K values/affinities can be reliably done. As commented in the text (lines 118-121/124-125/142-150) ligand 1 behaves differently from the other compounds, indicating a different binding mode which, in the case of Tel12, turns into the formation of a 1:2 berberine:G4 adduct.

  • Table 1: Please refrain from using word »Thermodynamic« whentalking only about melting temperatures and equilibriumconstants. Also, symbols representing physical quantities ormathematical variables should be written in italic type. How werethe error values calculated? Is [Tel12] concentration of singlestrand or concentration of bimolecular structure? Are all bindingstoichiometries calculated to bimolecular structure Tel12?

The word “Thermodynamic” was deleted and Table 1 heading was rephrased into “Table 1. Spectroscopic features, binding constants values and G-quadruplex melting temperature changes observed for the studied systems”. l, K and Tm were put in italic type. The errors in logK are ± SD on the triplicates, this is now explained in Table 1. All over the text Tel12 concentrations and stoichiometries are related to the bimolecular structure. This is now better emphasised (3.1 paragraph – materials and main text 2.1 paragraph).

  • Line 145: [ligand]/[Tel] = ca. 1: “ca.” is not explained
  1. stays for “circa”, approximately. So this to be clearer, it has been changed to [ligand]/[Tel] ≈ 1.
  • Figure 2: Why did authors decide to show titration curves atwavelengths of 342 and 349 nm? These values can't be found in table 1 (lmax). Graph with spectra has axis label »Negativeabsorbance«. Why? Differences in measured absorbances arethe same as photometric accuracy (0.002). The obtainedequilibrium constants are therefore questionable. What does itmean “% formation relative to B”, specifically what does “B” stand for? Experimental conditions are described as:“Experimental conditions: see Figure 1B and 2”. What do themean by Figure 1B and 2? This is Figure 2 and there are noexperimental conditions. At the bottom of panels A and B, thereare some additional graphs which are not described (“Bottompanels are the residuals” is not an appropriate description). Theauthors are showing absorbance spectrum and relevantdeconvolutions. They should also show measured spectra offree ligand to compare it with deconvoluted spectra. Legend should be displayed next to graphs.

Sorry to notice just now that Figure 2 was so hard to read. Unfortunately, it is what the software produces and it’s not easy (sometimes impossible) to correct its editing. However, we did our best to improve this figure, that is now moved to the Supporting Information. The titration curves shown at 342 and 349 nm are just one of the approximately many titration isotherms used by the software at each of the wavelengths of the uploaded spectra. These wavelengths were chosen only as an example and are just 1 nm far from the values reported in Table 1. Nevertheless, we agree that this is a misleading and not refined point of this figure: we changed in the new version the wavelengths to match exactly those cited in Table 1. “Negative absorbance” is a warning of the software in case some absorbances are found negative at the end of refinement and deconvolution procedure. However, the plot shows that this warning has negligible effect as the spectra of the different contributions are not noticeably under the zero line. This means that a few points are below zero for a so little amount, that the overall repercussion on the fit is negligible. Nevertheless, we agree that this is a misleading and not refined point of this figure: we improved it in the new version. Differences in absorption are not 0.002 but around 0.02 (see y-axis in the left panels). As for the reliability of the K values extracted, please see the previous reply. Given the simple 1:1 model found to describe the binding features, it is written by the software as A+B = AB where A = G4, B = berberine. Thus, % formation relative to B is the fraction of a species with respect to total berberine amount. This is now explained in the legend. “Experimental conditions: see Figure 1B and 2”: the recall to Figure 2 was a typo, the idea was to connect to the experimental conditions in Figure 1B. However, to make everything clearer, we reported all the conditions in the legend of the figure (now Figure 2S, SI) and improved the description at “methods”. “Bottom panels are the residuals” was changed into the more clear “Bottom panels are the residuals, i.e. the differences (experimental value) – (calculated value)”. The spectrum of the free berberine, in the deconvolution panel, is that in green. The spectrum of the free ligand cannot be directly overlaid to the software output in this figure, but the latter agrees with the free ligand spectra provided in the manuscript and with the features in Table 1.

  • Figure 3: Graphs are not aligned. Legends are missing. Red lineis not visible. Why are authors not showing melting curve for3+G4? What kind of model function was used to draw melting curves?

The alignment was improved. The legend (definitions of the symbols) was not in the panel but written in the bottom legend: it has been changed according to the request. The red line visibility was improved. We thank the reviewer and really apologise for the typo of 3+G4 missing: this is now added. The curve were drawn according to a sigmoidal fit, this is now said in the legend

  • Lines 160-164: This discussion could/should be extended andimproved (e.g. correlation between strengths of binding andlength of the alkyl chains, differences in equilibrium constants of Tel12 and Tel23…). Also, in some places bold numbers are used to describe ligands (1, 2, 3, …) in other places these numbersare not bold.

The discussion was improved. Typos on the absence of bold were checked and corrected all along the text.

  • Figure 4: Panel labels are different in all figures ((a), (A), A).They should be unified.

Panel labels were unified

  • Line 193: »The 4 ligand« should be replaced by »The ligand 4«

Done

  • Line 266: Why were only 2, 3 and 4 selected for Hirshfeld surface’s analysis? I guess the selection was not based on ability of derivatives to form crystals since only 2/Tel12 and4/Tel12 adducts gave crystals suitable for diffraction analysis.So, what was the criteria for selection of these three derivatives?

The Hirshfeld surface analysis is a method developed to analyze and compare crystal structures. We applied the method to all the crystal structures we were able to obtain for this series of ligands, i.e. the two crystal structures reported in the manuscript, i.e. the crystal structures of Tel12/2 and Tel12/4, and to the crystal structure of Tel12/3, which was previously obtained and published. In the Section 2.2 solid state studies, we do explicit reference to the already published Tel12/3 structure: “…Notably, water bridged or direct DNA/ligand H-bonds, involving a benzodioxole oxygen, were found also in the crystal structure of the 3/Tel12 adduct [35]”.

However, following the comments of Reviewer 3, the section 2.3. Hirshfeld surface’s analysis was significantly shortened and the details of the methods moved in SI.

  • Line 306: “The higher DTm value found in solution solution…”This should be rewritten to something similar to; “The higher DTm value observed in UV-melting experiment…”

Done

  • Line 324: Perhaps “orientation” should be used instead of“poses”?

In this context the two words are synonim. However, as requested by the reviewer “poses” was replaced by “orientation”

  • Line 328: Why »on the other hand«? Are results different or unexpected?

The expression “on the other hand” was removed.

  • Table 2: Benzodioxole should be written in one line

Done

  • Line 473: Word »long« should be used instead of »length«.

Done

  • Line 475: Why »on the other hand«? The sentence »In the case of adducts formed with Tel23, the binding is still favorable. « is enough

The expression “on the other hand” was removed

Reviewer 3 Report

The paper titled “Probing the efficiency of 13-pyridylalkyl berberine derivatives to Human Telomeric G-quadruplexes binding: spectroscopic, solid state and in silico analysis” by Bazzicalupi at al describes biophysical, X-ray crystallographic, and computational study of interactions between five berberine derivatives and two models of human telomeric DNA (Tel12 and Tel23). The derivatives contain pyridine linked to C13 of berberine by (CH2)n linker. The study explored n = 1-5 although binding of 3 to human telomeric DNA was already published (ref 35). Overall, the reviewer enjoyed reading the paper. The experimental work is interesting and the quality of experimental work is good. The reviewer has some comments about data analysis below. While the writing is clear, the paper requires careful proof reading.

Major comments:

·         The Conclusion in the paper brings all the work together. The reviewer would have liked to see a more detailed discussion as to how the linker affects binding and why; and how this work fits in with the published work on Berberine and its derivative. Are the reported ligands better, same or worse as compared to berberine and its other derivatives?  Similarly, why binding to Tel12 and Tel23 differ?

·         The resolution of both crystal structures is excellent but the Rwork/Rfree is rather high (28.2/31.0), this is especially true for 1.6 A structure. Could the authors verify that their structure is complete and provide some explanation as to why Rwork/Rfree are so high? Alternatively, the reviewer would have liked to see the pdb and mtz files to make that judgment for himself.

·         The authors need to explain what kind of model was used to fit their ligand binding UV data. All that the paper says is that the model with multiple equilibria was used which is not sufficient. Most of Figure 2 should go to SI section.

·         For higher accuracy, melting experiments for Tel23 could be repeated with lower amount of KCl present in the buffer to bring the melting transition in the middle of the graph. Currently it is at the very high temperature and the ligands’ effect is not well resolved.

·         The geometry of TATA tetrad needs to be presented, discussed, and compared to previous literature.

·         It is unclear why for the molecular modeling studies, parallel and hybrid structures of Tel23 were considered. The authors can run circular dichroism experiment to determine the conformation of Tel23 alone under their conditions and in the presence of ligands and then use that conformation for their modeling. In fact their data in ref 35 suggest that Tel23 adopts a hybrid (and not parallel) conformation alone and in complex with 3. Also, Figure 8 is too complicated to see anything.

·         In the Methods this statement “Ligands 1-5 (Scheme 1) were received by Naxospharma srl and used without further purification” is followed by the schematics of the synthetic procedure. Thus, it is unclear whether the authors or Naxospharma synthesized the ligands. Can you clarify?

·         The data for 3/Tel12 and 3/Tel23 are already reported in (35). This should be clearly stated throughout the paper and references should be provided (for example in Table 1 to both binding and melting data). Figures like S1C and S1G should be removed as this data is already in Ref 35, Figure 2).

Minor comments

·         P1 line 25 – what does n = 1 refers to?

·         Scheme 1: The compounds just differ by the linker length, so it would be better to have a single drawing of the compound with (CH2)n on it and state that n = 1 (NAX071), n = 2 (NAX120), etc. This will unclutter the text.

·         Some statements require references (p 2 line 79; p 3 line 97)

·         The compounds mostly refer to as 1, 2, 3, etc, but sometimes by their longer name of NAX071. Can you be consistent with your naming? Also, sometimes, 1, 2, 3 is bolded and other times it is not.

·         Figure formatting: Formatting of figure labels (A, B, vs a), b), … etc) is not consistent. Also in Figure 2 labels of axes are very small and hard to see. Lines in Figure 3 are hard to see. Some of them are too thick; some are too thin, red line is impossible to see.

·         Line 122-125 p 3-4 – the author argue that 2 produces adducts with the lowest affinity, but their data in Table 1 argues that both 1 and 2 have similar affinity.

·         Some of the Result should be moved to methods (e.g. line 135-137 p. 4; also lines 166-168).

·         Figures 5 and 6 could be combined for the ease of comparison. Both figures look too ‘messy’ and hard to read. The authors can make the ligand in thicker or darker wire and the bottom tetrads thinner or lighter.

·         Line 178-181 – how the angles between bases in tetrads were measured?

·         There are two paragraphs in the paper (lines 231-246) that only state that the structure of 2/Tel12 is the same as that of 4/Tel12. It would be more concise to make this statement (together with RMSD value) at the beginning of the crystallography section and describe the geometry and the organization of quatruplexes in both structures. The rest of the Results can then focus on a more interesting aspect that differ in both cases – ligand binding pocket.

·         The Hirshfeld surface’s analysis section (line 265-288) spends a lot of time explaining the method but is confusing and, in honesty, the reader still could not understand how it was performed (for example, this statement is very confusing: “This method is commonly applied to low molecular weight species, the overall structure of which is subjected to the calculation.” This section should either be rewritten for clarity and go Method section or be replace with relevant references.

·         Bottom part of Figure 7 seems to be redundant with the top part.

·         It is unclear what does this statement mean: “Melting experiments were carried out by heating the working solutions from 25°C to 90°C with a scan rate of 5°C/min every 6.5 minutes.” What was done every 6.5 min? Also, what concentration of samples were used and what was the buffer?

Author Response

  • The Conclusion in the paper brings all the work together. The reviewer would have liked to see a more detailed discussion as to how the linker affects binding and why; and how this work fits in with the published work on Berberine and its derivative. Are the reported ligands better, same or worse as compared toberberine and its other derivatives? Similarly, why binding toTel12 and Tel23 differ?

The Conclusion paragraph was improved as requested. Reference 61 was added.

  • The resolution of both crystal structures is excellent but theRwork/Rfree is rather high (28.2/31.0), this is especially true for1.6 A structure. Could the authors verify that their structure iscomplete and provide some explanation as to why Rwork/Rfreeare so high? Alternatively, the reviewer would have liked to seethe pdb and mtz files to make that judgment for himself.

Based on our experience, the R factor and Rfree in the case of DNA-quadruplex structures are somehow higher than the ones expected for a macromolecule, or at least compared to proteins. We do not have a definitive explanation for that. Nevertheless, our idea is that in general, while the quadruplex core is rigid and repeats in correct ordered manner, contributing to define the overall resolution, a high degree of disorder normally affects the more flexible loop and flanking residues. In the 2/tel12 and 4/tel12 structures, pseudosymmetry amplificates this effect. In fact, loop T6T7A8 has been found to assumes two different positions which are on conflict with their symmetry related copies. Author have explained this phenomenon in the experimental section “Each structure shows disordered TTA loops resulting in overall poor electron density maps. The A8 residue has been localized in two different positions with occupancy factor 0.5. For the base atoms of the T6 residue of 4/Tel12 0.0 occupancy factor was used, as they were not clearly localized in the map. The described disorder is most likely responsible for the tetragonal symmetry found, which can be interpreted as a pseudo-symmetry phenomenon. Nevertheless, all attempts carried out to treat these data with different cells and/or lower symmetry space groups gave no results.“ This sentence was not modified with respect to the original manuscript version.

  • The authors need to explain what kind of model was used to fittheir ligand binding UV data. All that the paper says is that themodel with multiple equilibria was used which is not sufficient.Most of Figure 2 should go to SI section.

Figure 2 is now in the SI (Figure S2). To clarify the type of fit the text now cites “according to different type of models related to multiple equilibria at different stoichiometric ratios” and some more detailed description is put in the “Methods”. Also, in the legend of figure 2 (now S2) is it explained: “Example of HypSpec2014 analysis of the absorbance changes observed upon addition of Tel12 to 1 according to a 1:2 model A + B = AB and 2A + B = A2B” for Figure S2(a) and “Example of HypSpec2014 analysis of the absorbance changes observed upon addition of Tel12 to 2 according to a 1:1 model A + B = AB” for Figure S2(b).

  • For higher accuracy, melting experiments for Tel23 could be repeated with lower amount of KCl present in the buffer to bringthe melting transition in the middle of the graph. Currently it is atthe very high temperature and the ligands’ effect is not well resolved.

We fully understand the reviewer’s concerns about the melting experiments on Tel23. However, it not easy to perform melting experiments with different amounts of KCl as this would crash with the need to use an homogeneous buffer all along the solution study and with the need to use conditions close to physiological ones and for whose the geometrical features of G-quadruplexes such as Tel12 and Tel23 are stable and well-known. The melting experiments were repeated in triplicate (this is now stated in the manuscript), and the DTm errors considers all contributions (on the single sigmoidal fit and over the triplicates). This results in high % error, as it should be (given the point arisen by the reviewer). However, we never use for data discussion the detailed numerical values and these errors are in our opinion not such to dampen the conclusion of these experiments: DTm values indicate a Tel23 G-quadruplex stabilisation which is present but significantly lower that what happens in the case of Tel12.

  • The geometry of TATA tetrad needs to be presented, discussed,and compared to previous literature.

Features of the TATA quartet have been described and compared with previous literature. Please see comment to point 1

  • It is unclear why for the molecular modeling studies, parallel and hybrid structures of Tel23 were considered. The authors can run circular dichroism experiment to determine the conformation of Tel23 alone under their conditions and in the presence of ligands and then use that conformation for their modeling. In fact their data in ref 35 suggest that Tel23 adopts a hybrid (and not parallel) conformation alone and in complex with 3. Also, Figure 8 is too complicated to see anything.

The Hybrid-1 DNA folding was chosen because it is the folding assumed in experimental conditions similar to those we used for solution studies. The structure was described in Dai, J. X.; Punchihewa, C.; Ambrus, A.; Chen, D.; Jones, R. A.; Yang, D. Z. Structure of the intramolecular human telomeric G-quadruplex in potassium solution: a novel adenine triple formation. NUCLEIC ACIDS RES, 2007, 35, 2440−2450 (ref .44 in the revised manuscript version. It was reference 55 in the original version). As temperature values measured in Melting Temperature experiments did not suggest variation of the DNA folding upon complexation, we decided to not perform additional CD experiments.

Calculation with the propeller folding were performed to compare modelling with X-ray structures and as additional evaluation of the accuracy of the method.

  • In the Methods this statement “Ligands 1-5 (Scheme 1) were received by Naxospharma srl and used without further purification” is followed by the schematics of the synthetic procedure. Thus, it is unclear whether the authors or Naxospharma synthesized the ligands. Can you clarify?

Synthesis was performed by the co-author Dr Paolo Lombardi at NaxosPharma. To be more clear we added the proper reference at the caption to CHART reporting the synthetic procedure.

  • The data for 3/Tel12 and 3/Tel23 are already reported in (35).This should be clearly stated throughout the paper andreferences should be provided (for example in Table 1 to both binding and melting data). Figures like S1C and S1G should be removed as this data is already in Ref 35, Figure 2).

The 3/Tel12 and 3/Tel23 systems have been completely re-analysed in the present work by new experiments. This was done so to ensure the complete homogeneity of the experiments and of data analysis procedure and, therefore, to ensure a reliable comparison of the numbers obtained for the set of 5 berberine derivatives considered in this work. Therefore, despite the consistency of the two (present and past) sets of data on 3, we are using in this paper only our newly and homogeneously obtained numerical values and findings. That’s why ref[35] is cited in the main text, but not in the Table. Even Figure S1C and S1G refer to the newly performed experiments

  • P1 line 25 – what does n = 1 refers to?

n = 1 is the length of the alkyl chain as defined in line 18

  • Scheme 1: The compounds just differ by the linker length, so it would be better to have a single drawing of the compound with (CH2)n on it and state that n = 1 (NAX071), n = 2 (NAX120), etc.This will unclutter the text.

We thank the referee for the suggestion. In the revised version report Sheme 1 was modified

  • Some statements require references (p 2 line 79; p 3 line 97)

References 1-3 have been added at p 2 line 79, while for p 3 line 97 the reference was already present in the old manuscript version (please check ref 32 at line 99

  • The compounds mostly refer to as 1, 2, 3, etc, but sometimes bytheir longer name of NAX071. Can you be consistent with yournaming? Also, sometimes, 1, 2, 3 is bolded and other times it is not.

Typos on the absence of bold were checked and corrected all along the text. The use of long names is a request by Naxospharma. The shorter numbering has been introduced for purpose of a more synthetic presentation and an easier reading. We prefer not to change the used names. We hope that the improvement of Scheme 1 may also use to easily let the reader find a definition of the two types of naming.

  • Figure formatting: Formatting of figure labels (A, B, vs a), b), …etc) is not consistent. Also in Figure 2 labels of axes are verysmall and hard to see. Lines in Figure 3 are hard to see. Someof them are too thick; some are too thin, red line is impossible to see.

The figure labels are now consistent. Figure 2 labels cannot be improved too much as they are screenshots of the software output; however, given that Figure 2 is now in the supporting information, this was divided into two panels, each one in a page, so to enlarge the dimensions and improve readability. Figure 3 was improved: all lines are 3 points width (the grey one was the only thicker being the reference, is it now like the others) and red line type was changed.  

  • Line 122-125 p 3-4 – the author argue that 2 produces adducts with the lowest affinity, but their data in Table 1 argues that both1 and 2 have similar affinity.

2 produces the lowest affinity in the set of compounds 2-5 that behave in the same way. 1 behaves differently, and the first binding constant may be similar to that of 2 but for the 1/Tel12 systems we have an additional “equilibrium relevant to 1:2 complex formation (as 1/Tel12 + Tel12  1(Tel12)2) is logK’ = 4.0 ± 0.2”. The sentence has been rephrased to make it clearer

  • Some of the Result should be moved to methods (e.g. line 135-137 p. 4; also lines 166-168).

In our opinion, the total removal of this very small introduction on the way data analysis is performed may let the subsequent reading of Table 1 non totally clear. Also, it clashes with the request of another reviewer. Therefore, we kept lines 135-137, but put also in the “methods” a more detailed description of the software for data analysis and relevant models.

  • Figures 5 and 6 could be combined for the ease of comparison.Both figures look too ‘messy’ and hard to read. The authors canmake the ligand in thicker or darker wire and the bottom tetradsthinner or lighter.

The two figures have been combined and now are Figure 4(a-d)

  • Line 178-181 – how the angles between bases in tetrads were measured?

The angular values have been measured using the Software Mercury, copyright of the Cambridge Structural Database and cited in ref 53 (ref 44 in the original version). The base/base dihedral angles were measured as plane/plane dihedral angles, each plane being the mean plane defined by the non hydrogen atoms of each single nucleotidic base in the tetrad.

  • There are two paragraphs in the paper (lines 231-246) that onlystate that the structure of 2/Tel12 is the same as that of 4/Tel12.It would be more concise to make this statement (together withRMSD value) at the beginning of the crystallography section and describe the geometry and the organization of quatruplexes in both structures. The rest of the Results can then focus on amore interesting aspect that differ in both cases – ligand bindingpocket.

We thank the referee for the suggestion. The section was significantly redreafted

  • The Hirshfeld surface’s analysis section (line 265-288) spends a lot of time explaining the method but is confusing and, in honesty, the reader still could not understand how it wasperformed (for example, this statement is very confusing: “This method is commonly applied to low molecular weight species,the overall structure of which is subjected to the calculation.”This section should either be rewritten for clarity and go Methodsection or be replace with relevant references.

Following the comments of the referee the description of the method has been moved to SI and the Section 2.3 significantly shortened.

  • Bottom part of Figure 7 seems to be redundant with the top part.

Figure 7 becomes figure 6 in the revised version. Bottom and top parts are only partially redundant. In the top part we used a matt depiction for the surfaces to give a clearer vision of the nucleic bases’ positions. In the bottom part we chose the transparent depiction, to allow an easier visualization of the ligands. The use of the transparent surfaces directly in the top figures resulted in not clear images, so we prefer not to change the figure

  • It is unclear what does this statement mean: “Meltingexperiments were carried out by heating the working solutionsfrom 25°C to 90°C with a scan rate of 5°C/min every 6.5minutes.” What was done every 6.5 min? Also, what concentration of samples were used and what was the buffer?

The statement was clarified and improved in “Melting experiments were carried out by heating the working solutions from 25°C to 90°C with a scan rate of 5°C/min, each step of 6.5 min being composed by 4.5 min rest, 1 min spectrum recording, 1 min temperature increase”. The concentrations and the buffer used were said in the legend of Figure 3 (now Figure 2). However, for an improved clarity and as requested by the reviewer, these information are now written also in the “methods” section as: “In these experiments [ligand] ≈ [G-quadruplex] = 1 – 2 × 10-5 M, buffer is KCl 0.1 M, NaCac 2.5 mM, pH 7.0”.

Round 2

Reviewer 1 Report

In the answer to the review of this referee, the authors said: "More than a theoretical computational investigation on the energetic contributions to the interaction, the study mainly focuses on the in solution characterizaton and on the structural aspects of the interaction of berberine derivatives with G-quadruplexes (G4). According to an approach typical of many medicinal chemistry studies that often uses empirical forcefields, our investigations mainly focuses on the forces that, overall, drive the stabilization of the complexes formed by the ligands and the G4. The detailed analysis of the electronic structure and energetics in the interaction is out of the aims of the present study"

For this referee if the investigations mainly focuses on the forces that, overall, drive the stabilization of the complexes formed by the ligands and the G4, as the authors say, the analysis of the energetics and different contributions that rule the affinity/selectivity of the studied compounds with GQ results of high importance since such energetics along with solvent effects will answer the question about the forces that, overall, drive the stabilization of the complexes formed by the ligands and the G4.

When the authors say "The geometrical criteria here usedfor the evaluation of intermolecular contacts are well established and accepted by the scientific community (cfr I. J. Bruno et al. J. Comput.-Aided Mol. Des., 1997, 11 , 525 -537, P. A. Wood et al., CrystEngComm(2013), 15, 65-72", the authors have to comment the shortcomings of this geometrical approach. QTAIM and NCI aproaches will always give a better evaluation for the intermolecular contacts than the geometrical approach since such QTAIM and NCI approaches already include implicitly a geometrical approach and, in addition, give the values of the electronic density or some isosurfaces that will give information about the strength of the weak interactions. In the case of the geometrical approach this strenght is only based on the length of the distances and less information is given. The authors have to comment this aspects.

When the authors say "When the target was DNA in hybrid 1 topology, the lignad was found to interact also with some loop or flankning residues. Involved residues are shown in figure 8 and kind of interactions are detailed in Table 2", in the current version figure 8 is not availabe.

When the authors say "We modified the Section 2.4. Molecular modelling study as follows: "To this aim, we chose the OPLS4 force field, which has proven to provide reliable results [43].", this reviewer went to reference 43 and this reference does not talk about the use and performance of the OPLS4 force field neither for G-quadruplexes nor for any kind of DNA. It is known in the bibliography that many efforts of authors of Barcelona and Olomouc aim at improving the conventional force fields used for regular duplex DNA for a better performance when studying G-quadruplexes. Parameters of the force fields imrpoved in Olomouc ara available.

When the authors say "Please, see point 18 in Reply to Reviewer 3", this Reviewer cannot see the reply to Reviewer 3.

Author Response

For this referee if the investigations mainly focuses on the forces that, overall, drive the stabilization of the complexes formed by the ligands and the G4, as the authors say, the analysis of the energetics and different contributions that rule the affinity/selectivity of the studied compounds with GQ results of high importance since such energetics along with solvent effects will answer the question about the forces that, overall, drive the stabilization of the complexes formed by the ligands and the G4.

When the authors say "The geometrical criteria here used for the evaluation of intermolecular contacts are well established   by the scientific community (cfr I. J. Bruno et al. J.Comput.-Aided Mol. Des., 1997, 11 , 525 -537, P. A. Wood et al.,CrystEngComm(2013), 15, 65-72", the authors have to comment the shortcomings of this geometrical approach. QTAIM and NCI approaches will always give a better evaluation for the intermolecular contacts than the geometrical approach since such QTAIM and NCI approaches already include implicitly a geometrical approach and, in addition, give the values of the electronic density or some isosurfaces that will give information about the strength of the weak interactions. In the case of the geometrical approach this strenght is only based on the length of the distances and less information is given. The authors have to comment this aspects.

We understand the points of Referee 1. We know the importance of computational studies performed by the Quantum Mechanical approach, and it is not our intention to reduce their impact. Nevertheless we have to underline, as we already did in our first reply letter, that our work is out of the context of the quantum-mechanics computing studies, but it is to be framed in the drug development studies’ context. As surely the Referee knows, drug development studies are normally based on several different techniques, each providing a piece of information, to be collapsed in the overall view of the studied molecules or targets. When molecular modelling is used, empirical forcefield methods are the first choice for medicinal chemists and SAR (structure-activity relationship) are deduced by applying geometrical criteria. Only as an exemplum, in the last ten years, the current three pharmaceutical companies with the highest turnover world wide (Johnson & Johnson, Pfizer and Roche) published 935 papers reporting molecular modelling data and out of these, only two reported QTAIM analyses.

We also fear that, despite the enormous effort required, the QTAIM analysis’ added value could be lost in the mass of data obtained with all the other techniques. As we already explained in our first reply letter, we would prefer to leave this type of study to a more theoretically focused work to be developed in the future.

When the authors say "When the target was DNA in hybrid 1topology, the lignad was found to interact also with some loop orflankning residues. Involved residues are shown in figure 8 and kind of interactions are detailed in Table 2", in the current versionfigure 8 is not availabe

We are not able to find any reference to figure 8 in the R1 version of the paper. Perhaps the reviewer refers to the sentence “Calculated models for the adducts formed with the type 1 hybrid structure agree with the results obtained by solution study. The structures are shown in Figure 6, and Table 2 summarizes the most important ligand/target interactions.” This sentence is at lines 315-317 in both R1 and R2 versions, and it correctly cites Figure 6.

When the authors say "We modified the Section 2.4. Molecularmodelling study as follows: "To this aim, we chose the OPLS4force field, which has proven to provide reliable results [43].", thisreviewer went to reference 43 and this reference does not talkabout the use and performance of the OPLS4 force field neitherfor G-quadruplexes nor for any kind of DNA. It is known in thebibliography that many efforts of authors of Barcelona and Olomouc aim at improving the conventional force fields used for regular duplex DNA for a better performance when studying G-quadruplexes. Parameters of the force fields improved in Olomouc are available.

We thank the referee for suggestions about the in house developed forcefield with parameters specifically improved for treating DNA G-quadruplex structures. We will test them in future works. In the current work we only used commercially available software from the Schrodinger company. In this framework, OPLS forcefields are well known and known to provide reliable results even with very different targets. Reference 43 is the original citation reference for the OPLS4, but Schrodinger’s web site reports as follows:

“Accurate parameters for proteins and nucleic acids

In addition to extensive coverage of small molecules, OPLS4 also contains improved parameters for proteins and nucleic acids, leading to marked improvement in structural stabilization during long MD simulations.”

However, our study also investigates the performace of the OPLS4 FF to reproduce the structures obtained by XRD analyses for ligands 2 and 4 in complex with the propeller G4 folding.

When the authors say "Please, see point 18 in Reply to Reviewer 3", this Reviewer cannot see the reply to Reviewer 3

We apologize for any inconvenience. In the following the Reply to Reviewer 3

“Following the comments of the referee the description of the method has been moved to SI and the Section 2.3 significantly shortened.”

Reviewer 2 Report

The authors have satisfactorily addressed all of my concerns. The only remaining concerns I have with the manuscript are
minor:

-Panel labels are still not unified (label position in Figure 2 is on the graph, position in other figures is bellow the graphs, sometimes only one parenthessis is used instead of two)

-in chapter "3.2 Solution studies" T should be in italic type

-perhaps instead of "CHART", "Scheme 2" should be used

Best regards.

Author Response

The only remaining concerns I have with the manuscript are minor:

-Panel labels are still not unified (label position in Figure 2 is on the graph, position in other figures is bellow the graphs,sometimes only one parenthessis is used instead of two)

Panel labels are now unified

-in chapter "3.2 Solution studies" T should be in italic type

Done

-perhaps instead of "CHART", "Scheme 2" should be used

We modified accordingly to the referee’s request
